# Isolation, Characterization and Antibacterial Activity of 4-Allylbenzene-1,2-diol from *Piper austrosinense*

**DOI:** 10.3390/molecules28083572

**Published:** 2023-04-19

**Authors:** Mengxuan Gu, Qin Wang, Rui Fan, Shoubai Liu, Fadi Zhu, Gang Feng, Jing Zhang

**Affiliations:** 1National Key Laboratory of Green Pesticide, Key Laboratory of Green Pesticide and Agricultural Bioengineering, Ministry of Education, Center for R&D of Fine Chemicals of Guizhou University, Guiyang 550025, China; 2Environment and Plant Protection Institute, Chinese Academy of Tropical Agricultural Science, Haikou 571101, China; 3Spice and Beverage Research Institute, Chinese Academy of Tropical Agricultural Sciences (CATAS), Wanning 571533, China; 4Key Laboratory of Genetics and Germplasm Innovation of Tropical Special Forest Trees and Ornamental Plants, Ministry of Education, Hainan Key Laboratory for Biology of Tropical Specific Ornamental Plants Germplasm, School of Forestry, Hainan University, Haikou 570228, China

**Keywords:** antibacterial activity, cell membrane, mechanism, physiological and biochemical index, *Xanthomonas oryzae* pathovar *oryzae*

## Abstract

Isolation for antibacterial compounds from natural plants is a promising approach to develop new pesticides. In this study, two compounds were obtained from the Chinese endemic plant *Piper austrosinense* using bioassay-guided fractionation. Based on analyses of ^1^H-NMR, ^13^C-NMR, and mass spectral data, the isolated compounds were identified as 4-allylbenzene-1,2-diol and (S)-4-allyl-5-(1-(3,4-dihydroxyphenyl)allyl)benzene-1,2-diol. 4-Allylbenzene-1,2-diol was shown to have strong antibacterial activity against four plant pathogens, including *Xanthomonas oryzae* pathovar *oryzae* (*Xoo*), *X. axonopodis* pv. *citri* (*Xac*), *X. oryzae* pv. *oryzicola* (*Xoc*) and *X. campestris* pv. *mangiferaeindicae* (*Xcm*). Further bioassay results exhibited that 4-allylbenzene-1,2-diol had a broad antibacterial spectrum, including *Xoo*, *Xac*, *Xoc*, *Xcm*, *X. fragariae* (*Xf*), *X. campestris* pv. *campestris* (*Xcc*), *Pectobacterium carotovorum* subspecies *brasiliense (Pcb*) and *P. carotovorum* subsp. *carotovorum* (*Pcc*), with minimum inhibitory concentration (MIC) values ranging from 333.75 to 1335 μmol/L. The pot experiment showed that 4-allylbenzene-1,2-diol exerted an excellent protective effect against *Xoo*, with a controlled efficacy reaching 72.73% at 4 MIC, which was superior to the positive control kasugamycin (53.03%) at 4 MIC. Further results demonstrated that the 4-allylbenzene-1,2-diol damaged the integrity of the cell membrane and increased cell membrane permeability. In addition, 4-allylbenzene-1,2-diol also prevented the pathogenicity-related biofilm formation in *Xoo*, thus limiting the movement of *Xoo* and reducing the production of extracellular polysaccharides (EPS) in *Xoo*. These findings suggest the value of 4-allylbenzene-1,2-diol and *P. austrosinense* could be as promising resources for developing novel antibacterial agents.

## 1. Introduction

The medicinal importance of plants has led to the exploration of plant extracts that are commonly used as antibacterial agents, because plant pathogenic bacteria are devastating to plants all over the world, which can cause various disease symptoms including spots, blights of leaf and soft rots of fruits [1,2,3]. Such events may greatly compromise the quality and output of crops [4,5]. In recent years, the occurrence of bacterial diseases in crops is increasing with the change in climate and planting structure. In some areas, bacterial diseases are becoming the predominant diseases and seriously limit the development of the agricultural industry. Currently, only a few varieties of pesticides, including kocide and thiadiazole copper (TC), have been registered for the control of bacterial diseases [6,7,8,9]. Given the practical issues that include huge losses led by bacterial diseases, the lack of targeted agents and the increasing resistance of pathogen strains are major concerns. Therefore, developing innovative antibacterial substitutes with safe and high-efficient attributes has been urgently required.

Plants are potential sources of natural bioactive compounds, many of which possess good antimicrobial activity and can be used as natural pesticides [10]. A large amount of research work has put a focus on searching for plant-derived fungicides. For example, some active compounds, such as physcion, osthole, and berberine, have been developed as botanical fungicides to effectively control plant diseases. Physcion is one of the common anthraquinone compounds that extensively exists in various plants. Increasing evidence suggested that physcion effectively inhibited the growth of phytopathogenic fungi and bacteria [7,11,12]. Osthole, a natural product derived from medicinal plants including *Cnidium monnieri* and *Angelica*, displayed multiple pharmacological actions such as immunomodulation and antimicrobial activity. Previous reports have shown that osthole inhibited the germination of spores and the growth of hyphae in *Sphaerotheca fuliginea*. Moreover, osthole derivatives have also shown superior controlled efficacy against *Phytophthora capsici* [13,14,15]. As a traditional Chinese medicine, berberine has a potent role in controlling plant disease and is also emerging as a promising botanical pesticide [6,16,17]. Currently, seeking novel plant-derived antibacterial drugs with high efficacy and low toxicity or identifying new properties using structural modifications has become the hot spot and provides strong challenge in the relevant research field.

*Piper* is an indispensable condiment with high economic value [18]. Multiple medicinal functions of *Piper* have also been confirmed [19], such as the relief in abdominal pain and diarrhea and protection of the liver. To date, many *Piper* species have exhibited a broad range of bioactivities, including antifungal, antibacterial and pesticidal properties [20]. *Piper austrosinense*, a peculiar genus of *Piper* plants in China, is distributed in Southern Chinese provinces, such as Hainan, Guangdong, Guangxi and Yunnan [21], the picture of *P. austrosinense* can be found in the Appendix A. Besides its well-known use as a culinary spice, *P. austrosinense* is mainly used as a medicine for treating toothache and traumatic injuries [22,23]. Liu et al., 1995 [24] separated nine compounds from *P. austrosinense*, two of which were identified as new amide alkaloids. However, the pharmacological effects of these amide alkaloids have not been confirmed yet. Chen et al., 2018 [25] separated eleven compounds (including protocatechualdehyde, protocatechuate, pipernonaline, etc.), among which pipernonaline displayed the inhibitory activity of butyrylcholinesterase. Moreover, the cytotoxic effects of the separated eleven compounds were evaluated in HepG2 liver cancer cells using MTT assays; the results showed that all compounds at a concentration of 30 μM did not exert cytotoxicity to the HepG2 cell line.

Our recent study observed that methanol extracts of *P. austrosinense* had significant antibacterial activity against *Xoo*. However, there is a lack of systematic investigation on the antibacterial activity of *P. austrosinense*. Therefore, in this study, we further identified the antibacterial properties of *P. austrosinense* based on the bioactivity-guided method while evaluating their antibacterial activities.

## 2. Results

### 2.1. Structural Elucidation of Isolated Compounds

In this study, two bioactive metabolites were successfully isolated from the methanol extract of *P. austrosinense* leaves and stems, using a tracing method of bioactivity. The isolated compounds were purified using column chromatography, whose structures were characterized using ^1^H NMR, ^13^C NMR and ESI-MS. Furthermore, their structures were confirmed by comparing with previously reported spectroscopic data [26,27]. Based on the spectroscopic data, the constituents were identified as 4-allylbenzene-1,2-diol (**1**), and (S)-4-allyl-5-(1-(3,4-dihydroxyphenyl)allyl)benzene-1,2-diol (**2**) (Figure 1).

### 2.2. In Vitro Antibacterial Activity

The bactericidal activities of 4-allylbenzene-1,2-diol and (S)-4-allyl-5-(1-(3,4-dihydroxyphenyl)allyl)benzene-1,2-diol are shown in Table 1. 4-allylbenzene-1,2-diol possessed excellent antibacterial activities against *Xac*, *Xoc*, *Xcm*, and *Xoo* at a concentration of 1000 μmol/L, with inhibition rates of 97.39%, 99.58%, 99.03%, and 99.24%, respectively, which was not significantly different from the positive control (kasugamycin, the structure can be found in the Appendix A) and was superior to that of (S)-4-allyl-5-(1-(3,4-dihydroxyphenyl)allyl)benzene-1,2-diol.

Due to significant antibacterial activity of 4-allylbenzene-1,2-diol, 4-allylbenzene-1,2-diol was selected for further analysis. The results of the 4-allylbenzene-1,2-diol bactericidal assay showed that 4-allylbenzene-1,2-diol inhibited the growth of all the tested strains with different degrees. In particular, among the tested bacteria, *Xoo*, *Xac* and *Xcm* were sensitive to 4-allylbenzene-1,2-diol in a range of 250–500 μmol/L. 4-Allylbenzene-1,2-diol completely inhibited the growth of the three different bacterial pathogens, with inhibition rates > 94% at the concentration of 500 μmol/L (Figure 2). At a low concentration of 250 μmol/L, 4-allylbenzene-1,2-diol still showed good inhibition against *Xac*, *Xoo* and *Xcm*, with inhibition rates of 72.28%, 56.72%, and 95.23%, respectively (Figure 2). The inhibitory efficiency of 4-allylbenzene-1,2-diol against *Xac*, *Xoo* and *Xcm* were equal to those achieved by kasugamycin at the concentration of 500 μmol/L (Figure 2). Higher inhibition activity of 4-allylbenzene-1,2-diol against *Xoo* and *Xcm* was obtained when the concentration decreased to 250 μmol/L.

### 2.3. Minimum Inhibitory Concentration (MIC)

The MIC of 4-allylbenzene-1,2-diol against 8 phytopathogenic bacteria is presented in Table 2. *Xoo* and *Xcm* were the most sensitive bacteria to 4-allylbenzene-1,2-diol, with MIC values of 333.75 μmol/L for both, which were lower than that of kasugamycin. The MIC values of 4-allylbenzene-1,2-diol against other bacteria ranged from 667.5 to 1335 μmol/L.

### 2.4. Growth Curve of the 4-Allylbenzene-1,2-diol against Xoo

The effects of various doses of 4-allylbenzene-1,2-diol on the growth curve of *Xoo* are shown in Figure 3. Compared with the control, the log periods were positively correlated with the treatment concentrations. Meanwhile, 4-allylbenzene-1,2-diol at MIC completely inhibited the growth of *Xoo* after treatment for 12 h.

### 2.5. In Vivo Bioactivity of 4-Allylbenzene-1,2-Diol against Xoo

The protective and curative effects of 4-allylbenzene-1,2-diol against *Xoo* are shown in Table 3 and Figure 4. 4-Allylbenzene-1,2-diol exhibited a strong protective effect against *Xoo* when the concentration was at 4 MIC (1335 μmol/L), with an efficacy of 72.73%, which was superior to the positive control kasugamycin (53.03%). In addition, the efficiency of 4-allylbenzene-1,2-diol decreased to 54.54% at 2 MIC (667.5 μmol/L), which was similar to that of the positive control kasugamycin. The in vivo curative activity of 4-allylbenzene-1,2-diol against rice bacterial leaf blight was weaker, with the efficacy of nearly 30% at 15 days after inoculation, which was significantly lower than that of kasugamycin (43.28%).

### 2.6. SEM Observation

The morphological changes of *Xoo* after treatments with various concentrations of 4-allylbenzene-1,2-diol are presented in Figure 5. Untreated *Xoo* cells (control) were rod-shaped with a relatively smooth surface and uniform in shape (Figure 5A), while the 4-allylbenzene-1,2-diol treatment for 5 h at MIC resulted in the appearance of ruffle in cells (Figure 5B). More apparent cell deformation was observed when the concentration of 4-allylbenzene-1,2-diol treatment increased to 2 MIC (Figure 5C). Moreover, *Xoo* cells treated with 4 MIC 4-allylbenzene-1,2-diol were deformed, collapsed, and wrinkled, while irregularly shaped holes were observed. (Figure 5D).

### 2.7. Membrane Permeability

The relative conductivity values constantly increased in control and 4-allylbenzene-1,2-diol-treated *Xoo* (Figure 6). Compared to the application of lower concentrations (1/4 and 1/2 MIC), the greatest efficacy of promoting the increases in relative conductivity was found in *Xoo* treated with 4-allylbenzene-1,2-diol at MIC and 2 MIC (Figure 6).

### 2.8. Cell Motility Assays

As shown in Figure 7, 4-allylbenzene-1,2-diol strongly inhibited the motility of *Xoo*, and the inhibitory effect significantly improved with increasing 4-allylbenzene-1,2-diol concentrations. Colony diameters of *Xoo* after 48 h of treatments with 1/4 MIC, 1/2 MIC, MIC and 2 MIC of 4-allylbenzene-1,2-diol were 22 mm, 12 mm, 1 mm and 0 mm, respectively, each of which was significantly less than that of the control.

### 2.9. Assay of 4-Allylbenzene-1,2-diol-Inhibited Biofilm Formation Assay

As illustrated in Figure 8, 4-allylbenzene-1,2-diol treatments at concentrations of 2 MIC, MIC, 1/2 MIC and 1/4 MIC resulted in reductions of biofilm formation by 83.92%, 56.64%, 44.98% and 31.50%, respectively, when compared to controls.

### 2.10. Extracellular Polysaccharide (EPS) Production

To examine the effect of 4-allylbenzene-1,2-diol on the production of EPS, *Xoo* was treated with multiple concentrations of 4-allylbenzene-1,2-diol. The results showed that 4-allylbenzene-1,2-diol at 2 MIC, MIC, 1/2 MIC and 1/4 MIC led to reductions of 74.01%, 46.49%, 32.84% and 5.35%, respectively, compared to controls (Figure 9).

## 3. Materials and Methods

### 3.1. Experimental Materials and Reagents

Leaves and stems of *P. austrosinense* were randomly collected in Xinglong Tropical Botanical Garden, Wanning City, Hainan Province, China (18°44′ N, 110°11′ E). The obtained plants were identified by the Spice and Beverage Research Institute, Chinese Academy of Tropical Agricultural Sciences. Eight phytopathogenic bacterial strains used for in vitro antibacterial screening included *Xoo* (bacterial leaf blight of rice), *X. oryzae* pv. *oryzicola* (*Xoc*, bacterial leaf streak of rice), *Xac* (citrus bacterial canker), *X. campestris* pv. *mangiferaeindicae* (*Xcm*, bacterial black spot of mango), *X. fragariae* (*Xf*, bacterial angular leaf spot of strawberry), *X. campestris* pv. *campestris* (*Xcc*, black rot of cabbage), *Pectobacterium carotovorum* subsp. *brasiliense* (*Pcb*, bacterial soft rot of potato) and *P. carotovorum* subsp. *carotovorum* (*Pcc*, bacterial soft rot of Chinese cabbage). All of the strains were incubated in 20% glycerol and preserved at −80 °C for further use. Strains were cultured on Luria-Bertani (LB) agar plates (containing 10 g of tryptone, 5 g of yeast extract, 10 g of NaCl, 16 g of agar and 1 L of distilled water) or in LB broth without agar at 28 °C in the dark. All the bacteria were obtained from the Environment and Plant Protection Institute of the Chinese Academy of Tropical Agricultural Science (Haikou, China).

### 3.2. Extraction and Isolation

The leaves and stems (2 kg) of *P. austrosinense* were air-dried and powdered, and then macerated in 10 L methanol for 7 days under room temperature. After filtration, the filtrate was evaporated under hypobaric condition to yield the crude extracts. The methanol extracts (320 g) were re-suspended in water (1.8 L) and partitioned with petroleum ether (3 × 1.8 L), followed by extraction with ethyl acetate (2 × 1.8 L). The petroleum ether and ethyl acetate solutions were evaporated under hypobaric conditions to produce the petroleum ether-extracted residues (43.85 g) and ethyl acetate-extracted (58.29 g). Preliminary experimental results demonstrated that the petroleum ether extract had potent antibacterial activity against *Xoo*, *Xoc*, *Xac* and *Xcm*. Petroleum ether extract was subjected to passing through a column of MCI gel (methanol-H_2_O: 70–100%) to obtain five components (Fr. A–Fr. E). Fr. C and Fr. D were separated on a column of Sephadex LH-20 column and eluted with methanol to obtain two fractions (Fr. CD1 and Fr. CD2). Afterward, Fr. CD1 was separated on a silica gel column by an elution with petroleum ether: ethyl acetate (15:1) to obtain compound **1** (30.0 mg). Fr. CD2 was applied to a silica gel column (petroleum ether/ethyl acetate =3:1) to yield compound **2** (14.3 mg). The characteristics of both compounds were identified as follows:

Compound **1**: (4-Allylbenzene-1,2-dio), colorless oil, ESI-MS: *m/z* 173 [M + Na]^+^, 149 [M − H]^−^; ^1^H-NMR (500 MHz, CDCl_3_) δH: 6.75 (lH, d, *J* = 7.8 Hz, H-5), 6.67 (IH, s, H-2), 6.57 (lH, d, *J* = 7.8 Hz, H-6), 5.88 (1H, m, H-8), 5.02(2H, m, H-9), 3.20 (2H, d, *J* = 6.4 Hz, H-7). ^13^C-NMR (125 MHz, CDCl_3_) δC: 133.4 (C-1), 115.9 (C-2), 143.6 (C-3), 141.8 (C-4), 116.2 (C-5), 121.2 (C-6), 39.5 (C-7), 137.8 (C-8), 115.7 (C-9). The ^1^H and ^13^C NMR data were in accordance with those of 4-allylbenzene-1,2-diol [27].

Compound **2**: ((S)-4-allyl-5-(1-(3,4-dihydroxyphenyl)allyl)benzene-1,2-diol), brown oil, ESI-MS: *m/z* 297 [M − H]^−^; ^1^H-NMR (500 MHz, CD_3_OD)δH: 6.65 (lH, d, *J* = 8.1 Hz, H-5′), 6.55 (lH, s, H-6), 6.55 (lH, s, H-3), 6.52 (lH, d, *J* = 2.1 Hz, H-2′), 6.42 (lH, dd, *J* = 8.1, 2.0 Hz, H-6′), 6.15 (1H, ddd, *J* = 16.9, 10.2, 6.3 Hz, H-8′), 5.85 (1H, m, H-8), 5.09 (1H, m, H-9′b), 4.95 (2H, m, H-9), 4.75 (1H, dt, *J* = 17.1, 1.8 Hz, H-9′a), 4.67 (lH, d, *J* = 6.3 Hz, H-7′), 3.16 (2H, m, H-7). ^13^C-NMR (125 MHz, CDCl_3_) δC: 130.4 (C-1), 134.2 (C-2), 117.3 (C-3), 144.2 (C-4), 144.4 (C-5), 118.0 (C-6), 37.4 (C-7), 139.4 (C-8), 115.3 (C-9), 136.3 (C-1′), 117.1 (C-2′), 146.0( C-3′), 144.5 (C-4′), 116.0 (C-5′), 121.2 (C-6′), 50.4 (C-7′), 143.1 (C-8′), 115.6 (C-9′). The ^1^H and ^13^C NMR date were in accordance with those of (S)-4-allyl-5-(1-(3,4-dihydroxyphenyl)allyl)benzene-1,2-diol [26].

### 3.3. In Vitro Antibacterial Bioassay

The antibacterial activities of 4-allylbenzene-1,2-diol against *Xoc*, *Xac*, *Xcm*, and *Xoo* were measured according to the method of Zhao et al., 2019 [28], with some modifications. Compounds **1** and **2** were dissolved in sterile distilled water with 1% acetone and diluted to a final concentration of 1000 μmol/L. Sterile distilled water with 1% acetone was used as the negative control, and Kasugamycin (Shanghai Macklin Biochemical Technology Co. Ltd., Shanghai, China) was used as the positive control. Approximately 10 μL of bacteria suspensions cultured on the phase of logarithmic growth was added to 190 μL LB containing tested compounds. The cultures were inoculated at 28 °C at 180 rpm for 12–18 h. The OD_600_ value was recorded to evaluate the bactericidal activity. Each treatment was repeated three times.

4-Allylbenzene-1,2-diol determination of the bactericidal spectrum was set up identically to that of the antibacterial activity assay in vitro. Antibacterial activities of 4-allylbenzene-1,2-diol on 8 phytopathogenic bacteria, including *Xoo*, were evaluated using turbidity assays. The 4-allylbenzene-1,2-diol was tested at final concentrations of 250 μmol/L and 500 μmol/L respectively. The OD value was measured after incubation to evaluate the antibacterial activity.

### 3.4. Determination of the Minimum Inhibitory Concentration (MIC)

The activities of 4-allylbenzene-1,2-diol against *Xoo*, *Xoc*, *Xac*, *Xcm*, *Xf*, *Xcc*, *Pcc*, and *Pcb* were examined by referring to the twofold dilution method, by which the minimum inhibitory concentrations were obtained [29]. The bacterial suspension (OD_600_ = 0.6) was added to the drug-contained medium to get the concentrations of 2670, 1335, 667.5, 333.75, 166.88, 83.44, 41.72, and 20.86 μmol/L, respectively. Kasugamycin at 1000, 500, 250, 125, 62.5, 31.25, 15.625, and 7.8125 μmol/L were used as positive controls, while 1% acetone was used as a negative control. Finally, the 96-well plate was incubated for 12 h at 28 °C in the incubator, and the lowest concentration was recorded as MIC when the blank control group became turbid. All measurements were repeated three times.

### 3.5. Growth Curve Assay

The effect of 4-allylbenzene-1,2-diol on the growth curve of *Xoo* was determined according to a previous method [30] with some modifications. 4-Allylbenzene-1,2-diol was dissolved in sterilized distilled water with 1% acetone and added to the culture medium to obtain final concentrations of 1/16 MIC, 1/8 MIC, 1/4 MIC, 1/2 MIC, and MIC. Sterilized distilled water with 1% acetone was used as a blank control. The bacterial suspension (OD_600_ = 0.6) was inoculated into LB medium-contained agents. Cell densities were monitored by measuring the optical density at 600 nm every 12 h during the cultivation of 84 h. All measurements were conducted in triplicate and means were considered.

### 3.6. In Vivo Antibacterial Activity against Xoo

The protective and curative activities of 4-allylbenzene-1,2-diol against rice bacterial leaf blight in potted plants were evaluated under greenhouse conditions. The experimental procedures followed the reference [31] with slight modifications. Rice seeds of ‘*Xiangliangyou 900*’ were germinated in the greenhouse and grown for 5 weeks. 4-Allylbenzene-1,2-diol in acetone was diluted with sterile distilled water (containing 0.1% Tween-20) to final concentrations of 2 MIC (667.5 μmol/L) and 4 MIC (1335 μmol/L). Sterile distilled water with 0.1% Tween-20 and 1% acetone served as blank controls, while kasugamycin (2% aqueous solution, 2000 μmol/L) was used as a positive control. For the protective activity experiment, the tips of rice leaves were cut using sterile scissors, and the *Xoo* in the logarithmic growth phase was inoculated at 24 h after evenly spraying 4-allylbenzene-1,2-diol solutions. In the curative activity experiment, the solutions were sprayed on the leaves at 24 h after inoculation. Additionally, the assay was repeated three times, with seven plants for each treatment. Inoculated rice plants were placed in a greenhouse at 70–80% relative humidity and 28 ± 2 °C. The disease index of the inoculated leaves was evaluated and photographed for 15 days. The degree of disease was graded as follows: level 0, no onset; level 1, lesions accounted for less than 1–5% of the leaf area; level 3, lesions accounted for 6–15% of the leaf area; level 5, lesions accounted for 16–25% of the leaf area; level 7, lesions accounted for 26–50% of the leaf area; level 9, lesions accounted for more than 50% of the leaf area. The disease index and control effect were calculated based on the following formula; see Equations (1) and (2) for details:(1)$$\mathrm{Disease}\;\mathrm{index}=\frac{\Sigma (the\;number\;of\;diseased\;leaves\;in\;each\;grade\times corresponding\;grade\;value)}{\left(total\;number\;of\;leaves\;investigated\times the\;highest\;disease\;grade\;value\right)}\times 100\%$$
(2)$$\mathrm{Control}\;\mathrm{effect}\;(\%)=\frac{disease\;index\;in\;the\;control-disease\;index\;in\;the\;treated\;group}{disease\;index\;in\;the\;control}\times 100\%$$

### 3.7. Scanning Electronic Microscope (SEM)

Sample preparation for scanning electron microscopy was carried out according to the method provided by Liu et al., 2021 [32] with slight modifications. The bacterial sus pensions (OD_600_ = 0.6) were washed three times with 0.1 mol/L phosphate buffer (pH 7.2) and resuspended. Thereafter, 4-allylbenzene-1,2-diol was added to the bacterial suspension to make the final concentrations reaching to MIC, 2 MIC and 4 MIC, respectively, and then the mixture was shaken at 180 rpm for 5 h at 28 °C. The cells were obtained after centrifugation at 5000 rpm for 5 min, and then washed three times with 0.1 mol/L PBS (pH 7.2). Subsequently, the bacterial cells were fixed to dehydrate with 2.5% glutaraldehyde at 4 °C for 12 h, and then washed 3 times with 0.1 mol/L PBS (pH 7.2), dehydrated with 30%, 50%, 70%, 80%, 90% and 100% ethanol solution for 15 min in sequence, and freeze-dried for 12 h. Finally, the samples were flattened and sprayed with gold. A SEM (Quorum Technologies, SC7620, East Sussex, UK) was used to observe the morphological change of the bacterial membrane.

### 3.8. Membrane Permeability

The permeability of the bacterial membrane was expressed in the relative electric conductivity that was measured using the method of Ernst et al., 2000 [33] with minor modifications. Bacteria were cultured at 28 °C until the logarithmic growth phase, followed by centrifugation at 2000 rpm for 20 min. The supernatant was discarded, and the cells were re-suspended with sterile water. The concentration of bacterial suspensions was approximately 10^8^ CFU/mL. Different concentrations of 4-allylbenzene-1,2-diol were respectively added to the bacterial suspension and incubated at 37 °C for 24 h. Conductivity was measured at 0, 2, 4, 6, 8, 10 and 24 h after additions of 4-allylbenzene-1,2-diol and recorded as L_1_. The conductivity of the mixture in boiling water for 10 min was recorded as L_2_. The cell membrane permeability was calculated using the following formula 3:(3)$$\mathrm{Membrane}\;\mathrm{permeability}\;(\%)=\frac{L_{1}}{L_{2}}\times 100\%$$

### 3.9. Bacterial Motility Assay

Bacterial motility was measured using a swimming assay according to the method of Di et al., 2008 [34]. Cultures of *Xoo* (OD_600_ = 0.6) were prepared. The LB solid medium containing 0.3% agar powder was heated in a microwave oven and boiled until completely dissolved. After cooling down to 40 °C, the reagent containing 4-allylbenzene-1,2-diol was added to the culture medium, with the final concentrations being 1/4 MIC, 1/2 MIC, MIC and 2 MIC, respectively. The overnight cultured bacterial suspension containing 4-allylbenzene-1,2-diol was drop-inoculated to the center of semisolid medium plates and incubated for 48 h at 28 °C. The bacterial solution without containing 4-allylbenzene-1,2-diol served as a blank group. The motility of bacterial cells was evaluated by measuring the diameter of the longest bacterial circles, and the measurement was repeated three times.

### 3.10. Assay of Biofilm Formation

The biofilm formation assay was performed based on the crystal violet staining method, as described by Du et al., 2018 [35] with slight modifications. The overnight cultured bacterial suspension (OD_600_ = 0.6) was inoculated into the LB medium containing 4-allylbenzene-1,2-diol whose final concentration was 1/4 MIC, 1/2 MIC, MIC and 2 MIC, respectively. For promoting the growth of biofilms, the mixed cultures in glass tubes were incubated at 28 °C for 5 d. After that, the culture medium was poured out and gently washed three times with distilled water. The cultures in each glass tube were stained with 2.5 mL of crystal violet (0.1%) for 30 min. After staining, the glass tubes were washed three times with 0.1 mol/L PBS (pH 7.2) to remove excess stains. Finally, the crystal violet-stained cells were solubilized with glacial 3 mL of glacial acetic acid. The absorbance of the biofilm was measured at 590 nm. Three replicates were performed.

### 3.11. Extracellular Polysaccharide (EPS) Production

EPS production was determined according to previous reports [22,36]. The bacterial cells were shakenly (180 rpm) cultured in LB media containing different concentrations (1/4 MIC, 1/2 MIC, MIC and 2 MIC) of 4-allylbenzene-1,2-diol for 72 h at 28 °C. Afterward, the cultures were centrifuged at 3000 rpm for 20 min, and the supernatants were collected. Finally, the supernatants were mixed with three-fold volumes of absolute ethanol and incubated overnight to precipitate EPS. The obtained EPS was pelleted via centrifugation and desiccation. The assay was performed three times.

### 3.12. Statistical Analyses

Data were presented as means ± standard deviations, and the data were subjected to variance analysis using the SPSS software (version 20.0, IBM Corp., Armonk, NY, USA). Significant differences between means were analyzed using Duncan’s Multiple Range Test at 0.05 levels. All assays for evaluating the activity of 4-allylbenzene-1,2-diol were performed with three replicates. The graphs were generated using Sigma Plot (version 12.5, Systat Software Inc., San Jose, CA, USA).

## 4. Discussion

In this study, (S)-4-allyl-5-(1-(3,4-dihydroxyphenyl)allyl)benzene-1,2-diol and 4-allylbenzene-1,2-diol were separated from *P. austrosinense* petroleum ether extract using the bioassay-guided assay. Structurally, (S)-4-allyl-5-(1-(3,4-hydroxyphenyl)allyl)benzene-1,2-diol is a dimer version of 4-allylbenzene-1,2-diol. Moreover, in vitro bioassays revealed that only 4-allylbenzene-1,2-diol possessed antibacterial activity. Few target bacteria have previously been used to screen active constituents of *P. austrosinense* for antibacterial activity, which might be one of the reasons why we failed to obtain more active ingredients. Additionally, the partial fractions obtained from column chromatographic separation were not further separated, which were likely to possess active components that could inhibit phytopathogenic bacteria.

4-Allylbenzene-1,2-diol is a simple phenolic compound that has been reported to have a variety of biological functions such as antifungal, anticancer and antioxidant properties [37]. Ali et al., 2010 [38] reported that 4-allylbenzene-1,2-diol is the most active compound extracted from *Piper betle* L. and has antibacterial properties, particularly for treating topical infections. According to a report by Sharma et al., 2009 [39], 4-allylbenzene-1,2-diol may be a promising compound that is developed as an antibacterial agent to treat oral diseases. In addition, 4-allylbenzene-1,2-diol was clinically confirmed to increase the sensitivity of bacteria to antibiotics due to its property of damage to the membrane in bacteria [40]. These findings indicate that 4-allylbenzene-1,2-diol is a promising functional compound. However, the bactericidal activity of 4-allylbenzene-1,2-diol against plant pathogens has not been reported. In our study, in vitro activity results showed that 4-allylbenzene-1,2-diol had good bactericidal activities against eight plant pathogenic bacteria, especially for *Xoo* and *Xoc*, with MIC values of 333.75 μmol/L, which were lower than that of kasugamycin. Kasugamycin has systemic activity and has been widely used to control disease in rice [41]. Thus, we used kasugamycin as the positive control to evaluated in vivo controlled efficacy of 4-allylbenzene-1,2-diol against *Xoo* on rice. 4-Allylbenzene-1,2-diol was found to have excellent protective activity, but its curative activity was relatively poor, which indicates that 4-allylbenzene-1,2-diol had a poor systemic activity or permeability in rice leaves, and its bactericidal activity might be induced by directly contacting with the pathogen. Furthermore, we explored the mechanism of bactericidal action 4-allylbenzene-1,2-diol in vitro. The results of SEM showed that 4-allylbenzene-1,2-diol caused the concaves and perforations in bacteria, which indicated that 4-allylbenzene-1,2-diol might disrupt cell membrane integrity. Moreover, 4-allylbenzene-1,2-diol promoted the increase in the relative conductivity of *Xoo* in a dose-effect manner, which further confirmed that this active compound could trigger damage to the cell membrane, resulting in the leakage of cellular contents. These findings indicated that the bactericidal mechanism of 4-allylbenzene-1,2-diol against plant pathogenic bacteria could be related to its damage to the cell membrane, which was consistent with the results obtained by Singh et al., 2021 [40].

Further results showed that 4-allylbenzene-1,2-diol limited the movement of *Xoo* in addition to inhibiting the growth of pathogenic bacteria. The swimming mobility of plant pathogenic bacteria is considered to be directly correlated with pathogenicity, and is also able to promote the formation of biofilm while helping the interaction between the bacterium and host, thus serving to enhance the infectious ability of the bacteria [42]. In this study, the swimming mobility of *Xoo* decreased by more than 50% when applied 4-allylbenzene-1,2-diol at the minimum concentration (1/4 MIC). Interestingly, 4-allylbenzene-1,2-diol showed a weaker inhibition on the growth of *Xoo* under the same conditions (1/4 MIC, 48 h), which indicated that the motility of *Xoo* could be more sensitive to 4-allylbenzene-1,2-diol. Tans-Kersten et al., 2001 [43] found that the loss of motility could significantly reduce the pathogenicity of *Ralstonia solanacearum*-caused bacterial wilt disease in tomato plants. In this study, the decreased infectious ability of *Xoo* due to the treatment of 4-allylbenzene-1,2-diol could be associated with the inhibition of swimming mobility. The formation of bacterial biofilms can enhance the tolerance of bacteria to bactericides, which increases the difficulty of preventing and controlling the infection of bacteria [44,45]. Exocellular polysaccharides, as one of the main constituents of biofilm, are closely related to the pathogenicity of bacteria of plant pathogens [46,47,48]. Chen et al. 2016 [49] reported that the natural product resveratrol inhibited the formation of biofilms of *Ralstonia solanacearum,* which contributed to the improvement of antibacterial ability. In our study, 4-allylbenzene-1,2-diol inhibited the biofilm and reduced the exopolysaccharide production of *Xoo*. Based on the above results, we speculated that the antibacterial effect of 4-allylbenzene-1,2-diol against *Xoo* might be associated with reduced pathogenicity by inhibiting polysaccharide synthesis and secretion, bacterial swimming mobility, and biofilm formation.

## 5. Conclusions

In this study, 4-allylbenzene-1,2-diol, an active compound was separated from the endemic plant *P. austrosinense* in China, which exhibited strong antibacterial activity against plant pathogenic bacteria with a broad spectrum. The antibacterial mechanism of 4-allylbenzene-1,2-diol might involve the loss of cell membrane integrity and reduced pathogenicity in plant pathogens. The results suggested that 4-allylbenzene-1,2-diol and the medicinal plant *P. austrosinense* could be potential sources of developing novel bactericides. Further studies will aim to elucidate the antibacterial molecular mechanisms as well as investigate their controlling efficiency in field conditions.

## Figures and Tables

**Figure 1 molecules-28-03572-f001:**
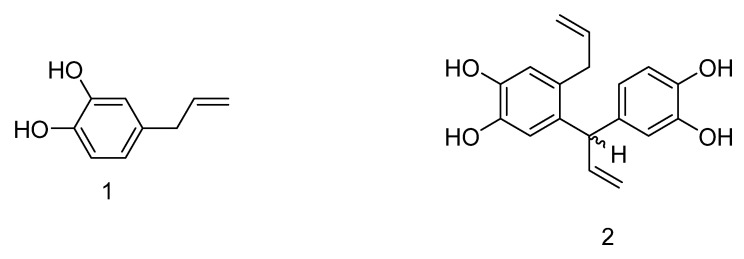
The chemical structures of the antibacterial active compounds from *Piper austrosinense*: 4-Allylbenzene-1,2-diol (**1**); (S)-4-allyl-5-(1-(3,4-dihydroxyphenyl)allyl)benzene-1,2-diol (**2**).

**Figure 2 molecules-28-03572-f002:**
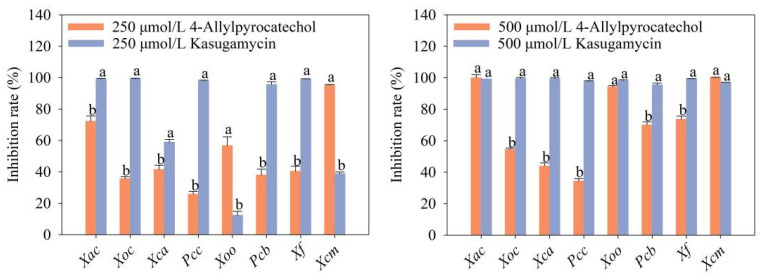
Antibacterial activity of 4-allylbenzene-1,2-diol against phytopathogenic bacteria. Note: Different lowercase letters in the same column show the significant difference at *p* < 0.05 level using Duncan’s new multiple range test. *Xac* represents *Xanthomonas axonopodis* pv. *citri*, *Xoc* represents *Xanthomonas oryzae* pv. *oryzicoia*, *Xca* represents *Xanthomonas campertris* pv. *campertris*, *Pcc* represents *Pectobacterium carotovorum* subsp. *carotovorum*, *Xoo* represents *Xanthomonas oryzae* pv. *oryzae*, *Pcb* represents *Pectobacterium carotovorum* subsp. *brasiliense*, *Xf* represents *Xanthomonas fragariae*, *Xcm* represents *Xanthomonas campestris* pv. *mangiferaeindicae*.

**Figure 3 molecules-28-03572-f003:**
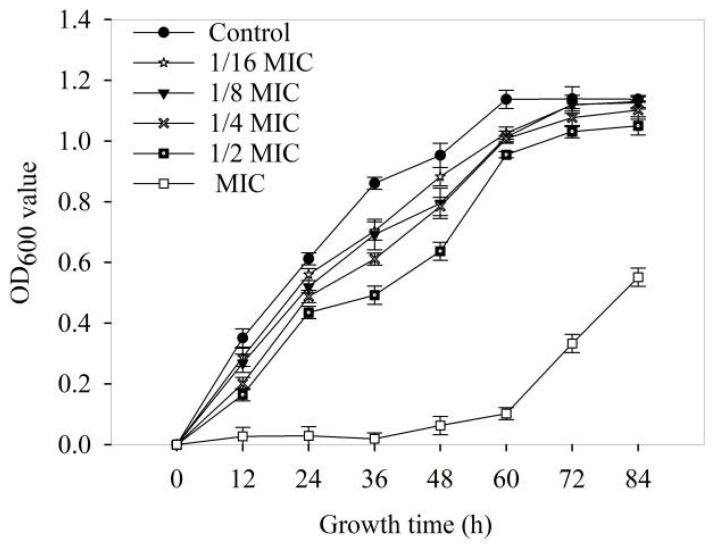
Effects of 4-allylbenzene-1,2-diol on the growth of *Xoo*.

**Figure 4 molecules-28-03572-f004:**
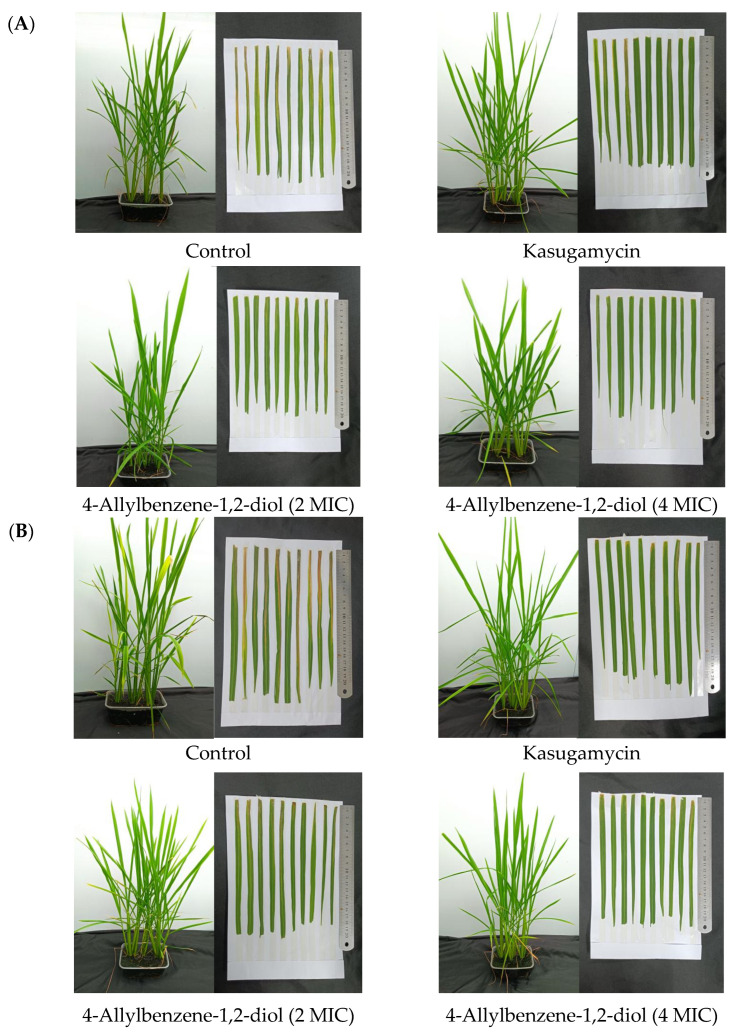
In vivo activities of 4-allylbenzene-1,2-diol against rice bacterial leaf blight (**A**) Protective activity; (**B**) Curative activity.

**Figure 5 molecules-28-03572-f005:**
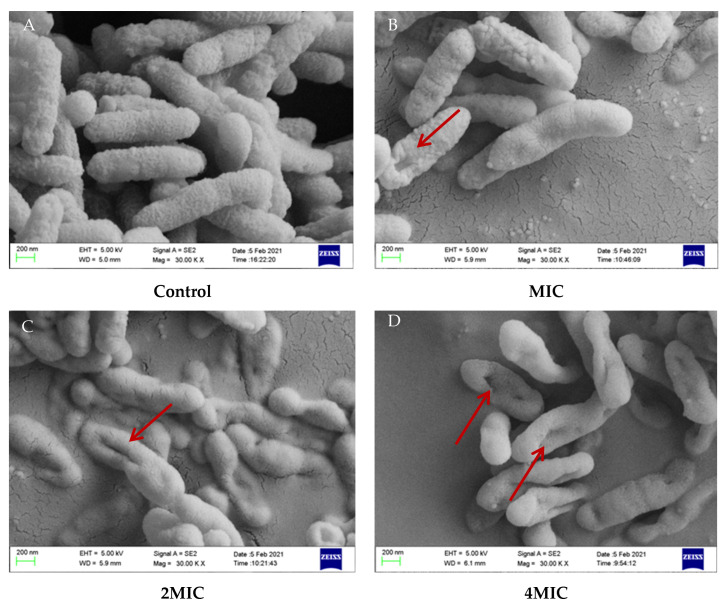
SEM image of *Xoo* after treatments with different concentrations of 4-allylbenzene-1,2-diol (**A**) control; (**B**) MIC; (**C**) 2 MIC; (**D**) 4 MIC.

**Figure 6 molecules-28-03572-f006:**
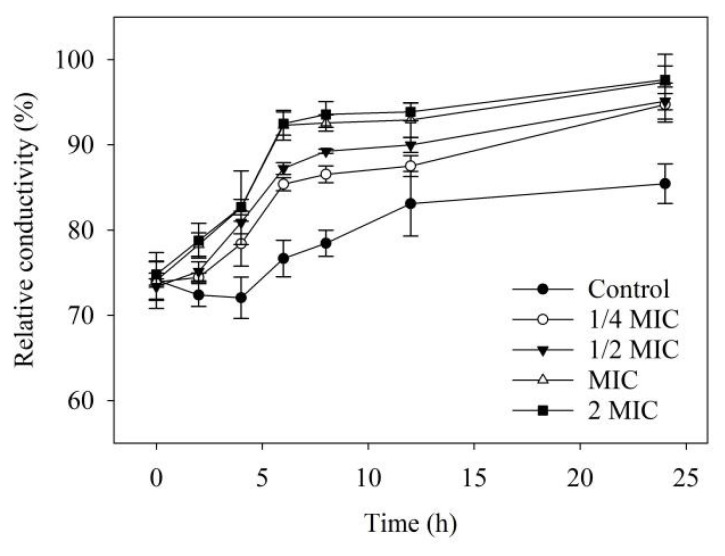
Effect of 4-allylbenzene-1,2-diol on the relative conductivity of *Xoo*.

**Figure 7 molecules-28-03572-f007:**
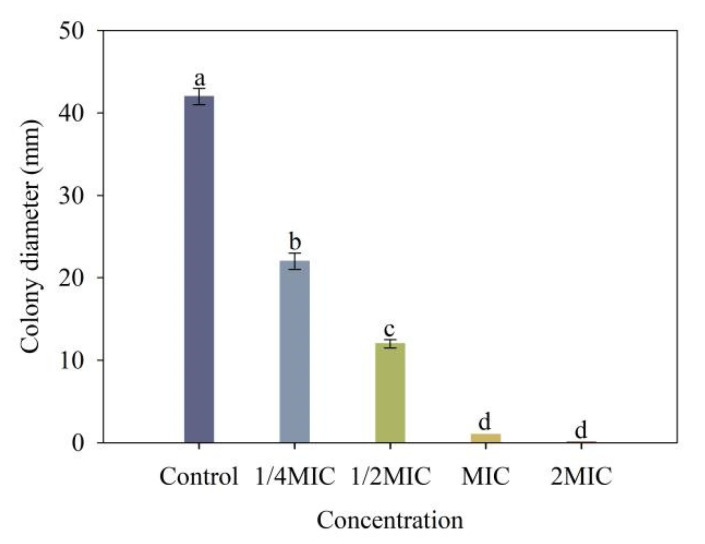
Effect of different concentrations of 4-allylbenzene-1,2-diol on swimming motility of *Xoo.* Note: Different lowercase letters show the significant difference at *p* < 0.05 level using Duncan’s new multiple range test.

**Figure 8 molecules-28-03572-f008:**
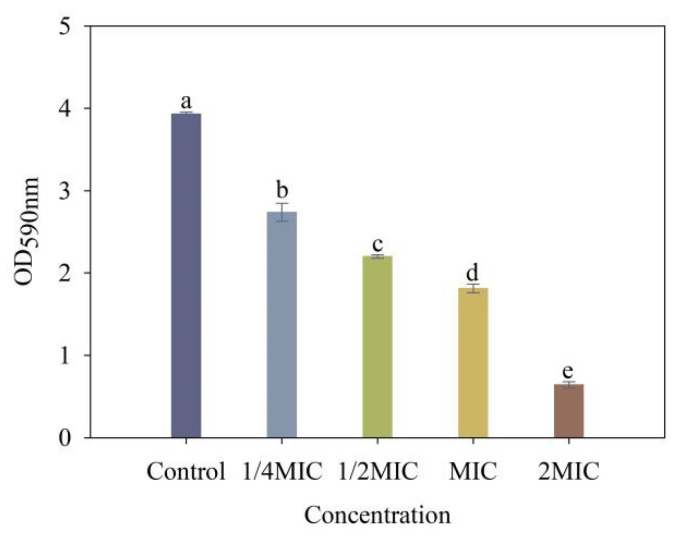
Effect of different concentrations of 4-allylbenzene-1,2-diol on biofilm formation of *Xoo.* Note: Different lowercase letters show the significant difference at *p* < 0.05 level using Duncan’s new multiple range test, respectively.

**Figure 9 molecules-28-03572-f009:**
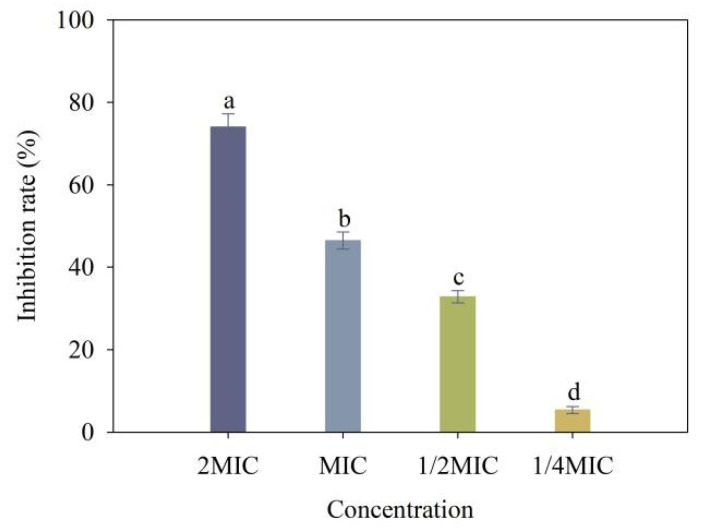
Effect of different concentrations of 4-allylbenzene-1,2-diol on EPS production of *Xoo.* Note: Different lowercase letters show the significant difference at *p* < 0.05 level using Duncan’s new multiple range test, respectively.

**Table 1 molecules-28-03572-t001:** Antibacterial activities of isolated compounds from *Piper austrosinens* against four phytopathogenic bacteria (concentration: 1000 μmol/L).

Strain	Inhibition Rate (%)		
	4-Allylbenzene-1,2-diol	(S)-4-Allyl-5-(1-(3,4-dihydroxyphenyl)allyl)benzene-1,2-diol	Kasugamycin
*Xac*	97.39 ± 0.17 a	24.73 ± 1.84 b	99.35 ± 0.25 a
*Xoc*	99.58 ± 0.96 a	37.53 ± 4.35 b	100.00 ± 0.22 a
*Xcm*	99.03 ± 0.24 a	40.44 ± 2.23 b	99.17 ± 0.50 a
*Xoo*	99.24 ± 0.05 a	30.97 ± 2.99 b	98.54 ± 0.25 a

Note: Different lowercase letters in the same column show the significant difference at *p* < 0.05 level using Duncan’s new multiple range test.

**Table 2 molecules-28-03572-t002:** Minimum inhibitory concentrations of 4-allylbenzene-1,2-diol against phytopathogenic bacteria.

Bacteria	Minimum Inhibitory Concentration (μmol/L)	
	4-Allylbenzene-1,2-diol	Kasugamycin
*Xac*	667.5	250
*Pcc*	1335	250
*Xcc*	1335	500
*Pcb*	1335	62.5
*Xoo*	333.75	500
*Xoc*	333.75	125
*Xcm*	333.75	500
*Xf*	1335	62.5

**Table 3 molecules-28-03572-t003:** Protective and curative activities of 4-allylbenzene-1,2-diol against *Xoo*.

Chemicals	Protective Activity (15 Days after Spraying)			Curative Activity (15 Days after Spraying)		
	Morbidity (%)	Disease Index (%)	Control Efficiency (%)	Morbidity (%)	Disease Index (%)	Control Efficiency (%)
4-Allylbenzene-1,2-diol (2 MIC, 667.5 μmol/L)	100	37.04 b	54.54 ± 6.38 b	100	60.49 b	26.86 ± 5.65 b
4-Allylbenzene-1,2-diol (4 MIC, 1335 μmol/L)	100	22.22 c	72.73 ± 5.60 a	100	56.79 b	31.34 ± 3.97 b
Kasugamycin (4 MIC, 2000 μmol/L)	100	38.27 b	53.03 ± 4.61 b	100	46.91 c	43.28 ± 7.83 a
Control	100	81.48 a	-	100	82.71 a	-

Note: Different lowercase letters in the same column show the significant difference at *p <* 0.05 level using the Duncan’s new multiple range test.

## Data Availability

Not available.

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
