# Peer review of "Isolation, Characterization and Antibacterial Activity of 4-Allylbenzene-1,2-diol from Piper austrosinense"

_molecules, 2023, doi:10.3390/molecules28083572_

Round 1

Reviewer 1 Report

Manuscript Number: molecules-2317274
Entitled: Isolation, characterization and antibacterial activity of 4-allylpyrocatechol from Piper austrosinense

This is an interesting scientific study. Therefore, the manuscript is suitable for Molecules after considering the below comments:

  1. Title. Possibly will be good to change it from “of 4-allylpyrocatechol from” into “of 4-allylpyrocatechol and neotaiwanensol B from”. The Authors discuss the influence of both molecules.
  2. In conclusion, should be information about isolated neotaiwanensol B. Please see the abstract part.
  3. Piper austrosinense; please add it’s a photo.
  4. Figure 1. Please unify both structures; the same rings, bond, etc. Please add the structure of kasugamycin from Table 1.
  5. Typos; page 3 is “4-allylpyrocatechol” should be “4-Allylpyrocatechol” because it starts a new sentence, and page 4 is the same. Page 11; is “4-allylpyrocatechol” should be “4-Allylpyrocatechol,”
  6. I suggest using in the abstract or experimental part UIPAC names for 4-allylpyrocatechol (4-allylbenzene-1,2-diol) and neotaiwanensol B ((S)-4-allyl-5-(1-(3,4-dihydroxyphenyl)allyl)benzene-1,2-diol).
  7. Page 9; 28 ℃, page 10; 2 ℃. please check if this is the right symbol for a degree and passim.
  8. References part. Please change “-” into “–” and check ref. 1. Pages 462-9, possibly should be 462–469.

Author Response

Q1. Title. Possibly will be good to change it from “of 4-allylpyrocatechol from” into “of 4-allylpyrocatechol and neotaiwanensol B from”. The Authors discuss the influence of both molecules.

Response: The neotaiwanensol B isolated from Piper austrosinense is essentially inactive, in the manuscript, we mainly discussed the mechanism of action of 4-allylpyrocatechol. Therefore, there is no neotaiwanensol B in the title.

Q2. In conclusion, should be information about isolated neotaiwanensol B. Please see the abstract part

Response: The neotaiwanensol B isolated from Piper austrosinense is essentially inactive, and our paper mainly investigated the mechanism of action of 4-allylpyrocatechol. So, there is no information about neotaiwanensol B in the conclusion part.

Q3. Piper austrosinense; please add it’s a photo.

Response: Thank you very much for your suggestion. Piper austrosinense pictures have been added to the Supplementary materials.

Q4. Figure 1. Please unify both structures; the same rings, bond, etc. Please add the structure of kasugamycin from Table 1.

Response: We sincerely thank you for your careful reading. We have re-drawn the structure diagram. In the revised version, Kasugamycin structure has been added to the Supplementary materials.

Q5: Typos; page 3 is “4-allylpyrocatechol” should be “4-Allylpyrocatechol” because it starts a new sentence, and page 4 is the same. Page 11; is “4-allylpyrocatechol” should be “4-Allylpyrocatechol,”

Response: Thank you very much for reminding. In the revised manuscript, the “4-allylpyrocatechol” has been replaced with “4-Allylpyrocatechol” 

Q6: I suggest using in the abstract or experimental part UIPAC names for 4-allylpyrocatechol (4-allylbenzene-1,2-diol) and neotaiwanensol B ((S)-4-allyl-5-(1-(3,4-dihydroxyphenyl)allyl)benzene-1,2-diol).

Response: Thank you very much for the constructive advice.According to your suggestion, we have changed it to UIPAC names.

Q7: Page 9; 28 ℃, page 10; 2 ℃. please check if this is the right symbol for a degree and passim.

Response: Thanks for the suggestion, and we have made modification using standard symbol of degree.

Q8: References part. Please change “-” into “–” and check ref. 1. Pages 462-9, possibly should be 462–469.

Response: We are sorry for our incorrect writing and have corrected it throughout the text.

Reviewer 2 Report

Comments and suggestions for authors are attached

The article entitled “Isolation, characterization and antibacterial activity of 4-allylpyrocatechol from Piper austrosinense is written however there are some minor revisions.

1.      Add some new keywords. The key words should not belong to title. Arrange the key words in alphabetical order.

2.      Abstract: Line 1: Replace the word “searching” by “Isolation”.

3.      Abbreviations must be complete in the first instance in abstract.

4.      Abstract: Revise and rephrase the abstract carefully.

5.      Introduction: Some sentences are too long. Shorten them in a meaningful way.

6.      Figure 2: Enlarge Figure 2 and improve its resolution. Also change the color for reader attraction.

7.      Add a paragraph about the medicinal importance of plant in the start of introduction section. You can read and get idea from these recent published papers; doi:10.3390/foods9030374;doi:10.3390/antibiotics9050226; 0.2174/1568026621666210701124628

8.      What does “Xoo” clarify it?

9.      4-allylpyrocatechol determination of the bactericidal spectrum was performed using the same method as for antibacterial the activity assay in vitro. Revise and rephrase this sentence.

10.  Formulas “Disese Index” “Control effect (%)”  membrane permeability etc should be reframed in equation form and assign them number like equation 1 ,equation 2, equation 3 etc.

11.  Figure 1: Add “chemical” before structures

12.  Table 1: Add “isolated” before compounds.

13.  NMR data.. Should be written as “NMR data”.

14.  All measurements were conducted in triplicate and means were calculated…Write “considered” instead of calculationed.

15.  Table 2: Standard deviations (SD) is missing in the data. Why ?

16.  As shown in Figure 9, treatments of different concentrations of 4-allylpyrocatechol suppressed the production of EPS to different degrees….Rephrase this sentence in a meaningful way.

17.  Figure 6-9: Change the color for reader attraction.

18.  2.1.structural elucidation…Add “of isolated compounds”

19.  2.6. SEM observation 4-allylpyrocatechol…Add “of” after observation.

20.  Scanning electronic microscope (SEM)…Add analysis  after SEM

21.  Assay of biofilm formation…Rephrase as “Biofilm formation assay”.

22.  Rephrase the conclusion section in a meaningful way.

23.  Revise the reference list according to journal style.

24.  The manuscript should be revised carefully. There are some minor grammatical mistakes throughout the manuscript.

25.  Check the Plagiarism of the manuscript.

Author Response

Q1. Add some new keywords. The key words should not belong to title. Arrange the key words in alphabetical order.

Response: Thanks for the suggestion. Several keywords including “physiological and biochemical index” and Xanthomonas oryzae pv. Oryzae” have been added. Also, we deleted the keywords “Piper austrosinense” and “4-allylpyrocatechol”.

Q2. Abstract: Line 1: Replace the word “searching” by “Isolation”.

Response: Thank you for your careful reading. As suggested, we have corrected “searching” to “Isolation”.

Q3. Abbreviations must be complete in the first instance in abstract.

Response: In the revised manuscript, we have corrected it.

Q4. Abstract: Revise and rephrase the abstract carefully.

Response: Thanks for your suggestion. We have carefully revised and rephrased the abstract.

Q5Introduction: Some sentences are too long. Shorten them in a meaningful way.

Response: Thanks for your suggestion. In the revised manuscript, we have shortened some sentences.

Q6. Figure 2: Enlarge Figure 2 and improve its resolution. Also change the color for reader attraction.

Response: Thanks for your suggestion. In the revised manuscript, we have enlarged Figure 2 for improving its resolution.

Q7. Add a paragraph about the medicinal importance of plant in the start of introduction section.

Response: Thanks for your suggestion. We have added some descriptions about the medicinal importance of plant at the start of the introduction. Moreover, the second paragraph of the introduction mainly described the antibacterial activity of medicinal plants.

Q8. What does “Xoo” clarify it?

Response: “Xoo” stands for Xanthomonas oryzae pv. oryzae, which is the abbreviation of Xanthomonas oryzae pv. oryzae that has been spelled out when it first appeared in the text.

Q9. 4-allylpyrocatechol determination of the bactericidal spectrum was performed using the same method as for antibacterial the activity assay in vitro. Revise and rephrase this sentence.

Response: According to your suggestion, we have corrected "4-allylpyrocatechol determination of the bactericidal spectrum was performed using the same method as for antibacterial the activity assay in vitro" to "4-Allylbenzene-1,2-diol determination of the bactericidal spectrum was set up identically to that of the antibacterial activity assay in vitro". 

Q10. Formulas “Disese Index” “Control effect (%)” membrane permeability etc should be reframed in equation form and assign them number like equation 1 ,equation 2, equation 3 etc.

Response: Thanks for your suggestion. In the revised manuscript, we have corrected it.

Q11. Figure 1: Add “chemical” before structures.

Response: Thanks for your suggestion. In the revised manuscript, we have corrected it.

Q12. Table 1: Add “isolated” before compounds.

Response: Thanks for your suggestion. In the revised manuscript, we have corrected it.

Q13. NMR data.. Should be written as “NMR data”.

Response: We feel sorry for our carelessness. In our resubmitted manuscript, we have corrected this mistake.

Q14. All measurements were conducted in triplicate and means were calculated…Write “considered” instead of calculationed.

Response: Thanks for your suggestion. In the revised manuscript, we have corrected it.

Q15. Table 2: Standard deviations (SD) is missing in the data. Why?

Response: The test was repeated three times and each MIC test result was the same value.

Q16. As shown in Figure 9, treatments of different concentrations of 4-allylpyrocatechol suppressed the production of EPS to different degrees….Rephrase this sentence in a meaningful way.

Response: As suggested by the reviewer, we have corrected “As shown in Figure 9, treatments of different concentrations of 4-allylpyrocatechol suppressed the production of EPS to different degrees” to “To examine the effect of 4-allylbenzene-1,2-diol on the production of EPS, Xoo was treated with multiple concentrations of 4-allylbenzene-1,2-diol”.

Q17. Figure 6-9: Change the color for reader attraction.

Response: Thanks for your suggestion. In the revised manuscript, we have corrected its color.

Q18. 2.1. structural elucidation…Add “of isolated compounds”.

Response: Thanks for your suggestion. In the revised manuscript, we have corrected it.

Q19. 2.6. SEM observation 4-allylpyrocatechol…Add “of” after observation.

Response: Thanks for your suggestion. In the revised manuscript, we have added it.

Q20. Scanning electronic microscope (SEM)…Add analysis after SEM. 

Response: Thanks for your suggestion. We have added analysis after SEM, including “and uniform in shape” and “while irregularly shaped holes were observed”.

Q21. Assay of biofilm formation…Rephrase as “Biofilm formation assay”.

Response: Thanks for your suggestion. In the revised manuscript, we have corrected it.

Q22. Rephrase the conclusion section in a meaningful way. 

Response: We have rephrased this part according to your suggestion.

Q23. Revise the reference list according to journal style. 

Response: Thanks for your suggestion. In the revised manuscript, the format of references has been adjusted according to journal style.

Q24. The manuscript should be revised carefully. There are some minor grammatical mistakes throughout the manuscript.

Response: Following your suggestion, we have consulted with an expert who carefully edited the writing. 

Q25. Check the Plagiarism of the manuscript. 

Response: According to the query report of the plagiarism-checking software, a 13.90 % of duplication ratio (excluding references) were reported, which should be in rational range.

Reviewer 3 Report

The manuscript in general is very well presented and written. There are only a few small observations that I think are important to clarify. Within the file they are marked in yellow and with comments.

Author Response

Q1. “strong antimicrobial activity” please review a cooment from disscusion.

Response: We thank you for your careful review. We have corrected the “strong antimicrobial activity” into “strong antibacterial activity”.

Q2. “active antibacterial activities” rearrange the text. It is not understandable.

Response: We were sorry for our careless mistake. Active antibacterial activities have been removed from the sentence.

Q3. What does CK mean?

Response: Thank you for your reminder. We have corrected the “CK” into “Control”.

Q4. “Compounds 1 and 2” Compound 2 was not active, but it is important to include these results in the discussion.

Response: Thank you for your suggestion. Our added content is as follows: Structurally, (S)-4-allyl-5-(1-(3,4-hydroxyphenyl)allyl)benzene-1,2-diol is a dimer version of 4-allylbenzene-1,2-diol. In addition, in vitro bioassays have shown that only 4-allylbenzene-1,2-diol possess antibacterial activity.

Q5. “4-allylpyrocatechol was found to have excellent protective activity, but its curative activity was relatively poor, which indicated that 4-allylpyrocatechol had a poor systemic activity or permeability in rice leaves, and its bactericidal activity may be caused by directly contacting with the pathogen.”This is not consistent with the general activity reported in the summary and conclusions. Please review.

Response: We thank you for your careful review. This section mainly discussed the possible reasons why protection is better than curation in vivo antibacterial activity against Xoo. The strong antibacterial activity was observed based on the bioassay of 4-allylbenzene-1,2-diol against a variety of plant pathogens in the abstract and conclusions. 

Q6The results of SEM showed that 4-allylbenzene-1,2-diol caused the 4-allylbenzene-1,2-diol concave and perforation in bacteria4-allylbenzene-1,2-diol could be removed from the sentence.

Response: Thanks for your suggestion. In the revised manuscript, we have removed it from the sentence.